# Vitamin D Receptor Gene Polymorphisms Associated with Childhood Autism

**DOI:** 10.3390/brainsci7090115

**Published:** 2017-09-09

**Authors:** Anna Cieślińska, Elżbieta Kostyra, Barbara Chwała, Małgorzata Moszyńska-Dumara, Ewa Fiedorowicz, Małgorzata Teodorowicz, Huub F.J. Savelkoul

**Affiliations:** 1Faculty of Biology and Biotechnology, University of Warmia and Mazury, 10-719 Olsztyn, Poland; anna.cieslinska@uwm.edu.pl (A.C.); elzbieta.kostyra@uwm.edu.pl (E.K.); ewa.kuzbida@uwm.edu.pl (E.F.); 2Regional Children’s Hospital in Olsztyn, 10-719 Olsztyn, Poland; basia.ch@interia.pl; 3Center for Diagnosis, Treatment and Therapy of Autism at the Regional Children’s Hospital in Olsztyn, 10-719 Olsztyn, Poland; magmoszyn@gmail.com; 4Cell Biology and Immunology Group, Wageningen University& Research, P.O. Box 338, 6700 AH Wageningen, The Netherlands; gosia.teodorowicz@wur.nl

**Keywords:** autism, vitamin D_3_, receptor, SNP analysis, polymorphism

## Abstract

Background: Autism spectrum disorder (ASD) is a group of heterogeneous, behaviorally defined disorders whereby currently no biological markers are common to all affected individuals. A deregulated immune response may be contributing to the etiology of ASD. The active metabolite of vitamin D_3_ has an immunoregulatory role mediated by binding to the vitamin D receptor (VDR) in monocyte, macrophages, and lymphocytes. The effects of vitamin D and interaction with the VDR may be influenced by polymorphism in the VDR gene. Methods: Genetic association of four different VDR polymorphisms (Apa-I, Bsm-I, Taq-I, Fok-I) associated with susceptibility to the development of autism in children was investigated. Results: We uniquely found an association between the presence of the *T* allele at position Taq-I and presence of the *a* allele at position Apa-I of the VDR gene with decreased ASD incidence. There was also an association between female gender and the presence of the *T* allele. We found no statistical significant correlation between VDR single nucleotide polymorphisms (SNPs) and vitamin D_3_ concentration in serum of ASD children. Conclusion: Genetic polymorphism in two SNP in VDR may be correlated with development of ASD symptoms by influencing functionality of vitamin D_3_ metabolism, while vitamin D_3_ levels were not significantly different between ASD and non-ASD children.

## 1. Introduction

Autism Spectrum Disorder (ASD) is a neurodevelopmental disorder characterized by abnormalities in social interactions, communication skills and restrictive or repetitive behaviors [1]. The prevalence of ASD ranges from 60 to 70 per 10,000 [2,3]. Its etiology remains unknown, but there is a clear promotion of the condition through a variety of factors including: genetics, autoimmunity, metabolic disorders and possible epigenetic modification through environmental or dietary factors [4,5,6].

Susceptibility to development of ASD has a profound genetic heritability and many genes and proteins have been implicated as causes of autism, but their individual contribution is still poorly understood. It has been suggested that mutations or single nucleotide polymorphisms (SNPs) in selected genes may influence the expression of those genes that are involved in brain development to generate a parsimonious hypothesis about brain dysfunctions that underlie autism [7,8].

One of the environmental factors potentially modulating brain development and its functioning is vitamin D [9,10]. Vitamin D_3_ deficiency during pregnancy and early childhood is suggested to be one of several risk factors for development of ASD in children that are genetically predisposed for autism [11,12,13]. Despite reports suggesting that children with autism had lower level of vitamin D_3_ [14,15], its precise role in development of the disease remains unclear.

Vitamin D_3_ (cholecalciferol), the natural form of vitamin D, is produced endogenously in the skin from sun exposure or obtained from diet [16,17]. Its active metabolite 1,25-dihydroxyvitamin D [1,25-dihydroxycholecalciferol or 1,25(OH)_2_D_3_], which acts through a specific vitamin D receptor (VDR), is involved in numerous processes such as transcriptional regulation of a large number of genes to mediate its genomic action on calcium homeostasis, ageing, immune response, immune modulation, cell proliferation and differentiation [18,19,20,21,22,23,24,25].

The VDR, a member of the nuclear steroid receptor family, is expressed in several human brain structures [26,27] and is therefore a suitable candidate gene for autism development [28]. VDR knockout mice showed severe impairments in the regulation of behavior [29,30], or hearing loss [31]. Important polymorphisms in the VDR gene were identified using different restriction enzymes in RFLP (Restriction Fragments Length Polymorphism) assays [21]. Polymorphisms detected by restriction enzymes Apa-I A/a (adenine->cytosine) (rs7975232) [32] and Bsm-I B/b (adenine -> guanine) (rs1544410) [33] were found in intron 8. Taq-I (rs731236) polymorphism was observed in exon 9, T/t (cytosine -> thymine), as a silent mutation [34], and missense Fok-I (rs2228570) polymorphism in exon 2 is connected with a change in the length of the protein resulting in a protein with 427 amino acids (f (thymine) variant), and with 424 amino acids (F(cytosine) variant). The second variant results in a more active form of the protein in some cell types [35,36,37].

VDR receptor gene polymorphisms were identified in various diseases as shown in wide reviews of Valdivielso and Fernandez [21] and Uitterlinden et al. [37]. Polymorphisms in Bsm-I, Taq-I, Apa-I and Fok-I were associated with renal diseases, cancer, neurolithiasis and diabetes. In addition, some authors [38,39,40,41,42] showed a correlation between VDR gene polymorphisms and susceptibility to asthma and atopic dermatitis. Abnormalities in the vitamin D receptor, and low levels of vitamin D were both linked with Parkinson’s disease [43,44].

In this study, we investigated the genetic association of four different VDR polymorphisms (Apa-I, Bsm-I, Taq-I, Fok-I) with susceptibility to the development of autism in children. We also correlated vitamin D_3_ concentration in serum with VDR polymorphism. When a positive correlation was established, it could be traced to the relevant SNP as a marker of the disease. In those conditions where the SNP had an impact on the body’s response to the actual vitamin D level, a personalized supplementation for the carriers of this particular SNP could be initiated.

## 2. Materials and Methods

### 2.1. Control and Patient Characteristics

Peripheral blood was collected from 108 children clinically diagnosed with autism spectrum disorder (ASD, ICD-F84) (91 male, 17 female, mean 6.8 years, range 3–11 years (ASD group). These patients were recruited in the Center for Diagnosis, Treatment and Therapy of Autism at the Regional Children’s Hospital in Olsztyn, Poland. Diagnosis was based on the International Classification of Diseases and Mental Disorders Behaviors ICD-10 as ICD-F84.0 [3], which identifies children on the basis of an interdisciplinary differential diagnosis including psychiatric examination (excluding mental illness) and evaluating cognitive intellectual parameters in the respondents (Leiter scale—standard IQ from 70 to 107; Wechsler—standard IQ from 90 to 104). Further ASD-patient identification included speech therapy evaluating speech development, EEG neurological examination, atypical pupillary light reflex evaluation, passive and participatory observation lasting from 6 to 12 months. From these patients also other relevant documentation, including names of parents and the opinions of educational institutions, and finally, video footage were evaluated. In addition, a group of 196 non-ASD children (98 male, 98 female, mean 8.5 years, range 4–18 years) was recruited from emergency department of Regional Children’s Hospital with no history of behavioral disorders (Control group). ASD and non-ASD groups included only children without vitamin D_3_ supplementation. The Local Bioethics Committee (number 19/2016 from the date of 18 May 2016) of the Hospital approved this study and informed consent was obtained from all children’s parents.

### 2.2. Polymorphism of VDR Genes in ASD and non-ASD Children with Autism

DNA was isolated from peripheral blood using GeneJET™ Whole Blood Genomic DNA Purification Mini Kit (Thermo Scientific, Waltham, MA, USA) according to the manufacturer’s instructions. Bsm-I, Taq-I, Apa-I and Fok-I VDR polymorphisms were assessed by polymerase chain reaction-restriction fragment length polymorphism (PCR-RFLP). DNA samples were used from those children that enabled us to type all four selected SNPs simultaneously. Names and notation of the alleles are widely used and taken from literature [21,32,33,34,45]. Primers examining the polymorphism in Fok-I were as previously described [45,46] with slight modifications, while primers for Bsm-I, Taq-I and Apa-I were designed with the Primer3 application [47]. The primer specificity was verified with the BLAST algorithm and primer sequences used for amplification of Fok-I, Bsm-I, Taq-I and Apa-I restriction enzyme polymorphisms are listed in Table 1.

PCR amplification was conducted in a thermal cycler according to the following program: initial denaturation: 94 °C for 3 min, proper denaturation: 94 °C for 30 s, attaching the starters at 61 °C for all genes for 30 s, synthesis: 72 °C for 30 s, final synthesis: 72 °C for 5 min, number of cycles: 40, cooling: 4 °C. The mixture in the volume of 25 μL consisted of DreamTaq™ Green Master Mix (Thermo Scientific, Waltham, MA, USA), specific primers, the DNA matrix, and ultrapure water (Sigma-Aldrich, Saint Louis, MO, USA). The yield and specificity of PCR products were evaluated by electrophoresis in 1.5% agarose gel (Promega, Fitchburg, MA, USA) and staining with GelGreen Nucleic Acid Gel Stain (Biotium, Hayward, CA, USA). Amplified fragments were digested with the appropriate restriction enzyme (Thermo Scientific, Waltham, MA, USA) (Table 1) according to the manufacturer’s instructions and visualized on a 2.5% agarose gel (Figure 1A–D). DNA sequencing of random chosen samples after amplification was used to confirm proper genotyping.

### 2.3. Vitamin D_3_ Concentration

The study included randomly selected 41 children (19 ASD, and 22 non-ASD) from whom peripheral blood was collected. We analyzed serum for vitamin D_3_ content in duplicate with a Human Vitamin D_3_ ELISA kit suitable to for serum and plasma, according to the manufacturer’s instruction (QAYEE-BIO, Shanghai, China). The kit is a double-antibody (containing two different 25-dihydroxyvitamin D-specific antibodies) sandwich ELISA with the detecting antibody conjugated to horseradish peroxidase (HRP). The same volume of liquid was placed in each microtiter plate well in all steps throughout the procedure and incubation steps were carried out for 60 min at 37 °C with gentle shaking (250 rpm) in microplate incubator (SkyLine ELMI Shaker DTS-4, Riga, Lithuania). The standard curve was tested in a concentration range of 0–200 ng/mL, while serial dilutions using 50 µL serum were used to determine the vitamin D_3_ concentration. Then, plates were washed five times and chromogen solution was added. After 15 min of incubation, 50 µL of stop solution was added. The absorbance was measured at the wavelength of λ = 450 nm using an ELISA reader (BiogenetAsys UVM 340, Cambridge, UK). The results were analyzed in the GraphPad Prism 4 application (v 6.01; GraphPad, San Diego, CA, USA).

### 2.4. Statistical Analysis

The genotype distribution among subjects was analyzed for Hardy-Weinberg equilibrium (HWE) using the chi-square test, and genotype and SNP allele frequencies were compared in ASD patients and non-ASD control groups by Fisher’s test. Odds ratios (ORs) and 95% confidence intervals (CIs) were calculated using logistic regression analysis and used to compare both allele frequencies in non-ASD controls and ASD patients, and allele frequencies between females and males. The risk of ASD development was estimated via wild-type genotype versus the wild/mutant and mutant-type genotypes. Vitamin D_3_ concentration results have been presented as a mean ± SEM. The mean values in non-ASD control and ASD groups were compared using Student’s *t*-test. Statistical analysis was conducted on GraphPad Prism software, with≤ 0.01 *p*-value considered statistically significant.

## 3. Results

### 3.1. Polymorphism of VDR Genes in non-ASD Control and ASD Children

The observed genotype frequencies of Fok-I (rs2228570), Bsm-I (rs1544410), Taq-I (rs731236), and Apa-I (rs7975232) polymorphisms in VDR in 196 of controls and 108 patients with ASD conformed to the Hardy-Weinberg equilibrium. This suggests no unexpected population stratification and no sampling bias.

At the VDR gene polymorphic site Fok-I (rs2228570); the 3 FF, Ff and ff genotypes were identified with 0.58 *F* allele frequencies in the entire research population (control and ASD groups). Of the total 304 participants, 106 had genotype FF, 143 had Ff, and 55 had ff.

At the Bsm-I (rs1544410) VDR gene polymorphic site the frequency of alleles *B* and *b* were determined in our study’s non-ASD control and in ASD diagnosed children. Three genotypes (BB, Bb, bb) were identified in the whole study population (control and ASD), with allele *B* frequency of 0.42. Of the total 304 participants, 36 had genotype BB, 185 had Bb and 83 had bb.

At the VDR gene polymorphic site Taq-I (rs731236); the 3 TT, Tt and tt genotypes were identified with 0.65 *T* allele frequencies in the entire research population (control and ASD groups). Of the total 304 participants, 125 had genotype TT, 146 had Tt, and 33 had tt.

At the VDR gene polymorphic site Apa-I (rs7975232); the 3 AA, Aa and aa genotypes were identified in the study population (control and ASD groups), with *A* allele frequency of 0.48. From all our control and ASD-affected children, 48 had genotype AA, 192 had Aa, and 64 had the aa genotype. Alleles *T* and *a* appeared two times more common in control, while the alleles *t* and *A* are more common is ASD. In ASD we reported 36% TA, 30% Ta, 20% tA and 14% ta, while in the non-ASD children we found 34% TA, 42% Ta, 8% tA and 16% ta.

Table 2 and Table 3 show the genotype distributions, allele frequencies and associations between genotype and ASD incidence at the 4 SNPs in VDR gene polymorphic site in non-ASD control and patients with ASD, and association with gender. These results suggested an association between the presence of the *T* allele at position Taq-I (rs731236) (OR = 2.08, 95% CI: 1.40–3.01, *p* < 0.0004), and the presence of the *a* allele at the position Apa-I (rs7975232) (OR = 2.65, 95% CI: 1.6–4.3, *p* = 0.0001) of the VDR gene under conditions of an decreased ASD incidence.

### 3.2. Vitamin D_3_ Concentration

Average vitamin D_3_ concentration in ASD group was 49.4 ng/mL (19.9–72.5 ng/mL; SDE = 4.23), and in Control group 41.5 ng/mL (22.2–70.9 ng/mL; SDE = 3.39) with no statistical difference. There were also no statistical significant differences according to VDR SNPs (Apa-I, Taq-I, Bsm-I, Fok-I) and vitamin D_3_ concentration in serum of the ASD group (Table 4). Vitamin D_3_ concentration was not correlated with VDR genotype in Control group because of too small number of alternative genotypes.

## 4. Discussion

Causes of autism are widely described to be multifactorial and include both genetic and environmental factors [12,48]. Epigenetics therefore plays an important role in ASD etiology and integrates genetic and environmental influences to deregulate neurodevelopmental processes. Nowotny et al. [49] showed that SNP allelic association methods are powerful tools in the identification of genetic factors that predispose to most common diseases. Therefore, single nucleotide polymorphism analysis in autism can identify risk factors for this disease by their presence as genomic markers. Earlier work linking autism with vitamin D deficiency due to skin color [50] and also during fetal development are not fully consistent but do suggest a possible effect of sunlight exposure at the time of birth [51]. These conflicting data suggest the presence of a genetic factor that can be identified using SNP analysis, and which is responsible for the deficiency of vitamin D [52,53]. In this study, we used SNP analysis to identify the differences between ASD and non-ASD controls in the distribution of Apa-I, Taq-I, Bsm-I and Fok-I genotypes in the VDR gene (Table 2 and Table 3), which plays an important role in the development and functioning of the brain.

The frequency of alleles and the distribution of genotypes in non-ASD control and ASD children for Bsm-I (0.42 for *B*), Apa-I (0.48 for *A*), and Fok-I (0.42 for *f*) was similar with data presented by Uitterlinden et al. [37], where in Caucasians polymorphisms are 0.34 of *f* in Fok-I, 0.42 of *B* in Bsm-I, 0.44 of *A* in Apa-I. In our research only the polymorphism in Taq-I was increased (0.65 in *T*) compared to the value of 0.43 for *T* as noted before. Our results uniquely suggest an association between the presence of the *T* allele at position Taq-I (rs731236) (OR = 2.08, 95% CI: 1.4–3.0, *p* < 0.0004), and presence of the *a* allele at position Apa-I (rs7975232) (OR = 2.65, 95% CI: 1.6–4.3, *p* = 0.0001) of the VDR gene and decreased ASD incidence. Our results suggest that the tA allele combination is 2.5 times more frequent in ASD children than in non-ASD children, despite the fact that these data need now to be confirmed in larger groups.

We also found an association between female gender and the presence of allele *T* in children (OR = 1.76, 95% CI: 1.2–2.6, *p* = 0.0035), which is consistent with data that autism is more common among boys [54], which have less frequent *T* allele (OR = 0.57, *p* = 0.0035) (Table 3). The number of female ASD children in this study (17 in 108 ASD children), however, was too low to allow a separate study on the influence of gender, beyond our reported correlation. Our results indicate that there was also no statistical correlation between the occurrence of the alleles at the polymorphic site of Fok-I and Bsm-I in VDR gene and susceptibility to ASD (Table 2). So far, only one study of Coskun et al. [55] showed correlation between Fok-I TT (ff), Taq-I CC (tt) and Bsm-I AA *(BB)* genotype and ASD. The polymorphism in the VDR receptor was also linked to childhood temporal lobe epilepsy, and several renal, skin, and bone diseases [21,37,45,56].

Moreover, our results suggest the lack of correlation between Apa-I, Taq-I, Bsm-I and Fok-I VDR polymorphisms and vitamin D_3_ concentration in serum. In both groups, ASD and Control, the results showed high disproportions in vitamin D_3_ concentration (19.9–72.5 ng/mL), and mean value was not statistically significant (49.4 ng/mL and 41.5 ng/mL, respectively, Table 4). Coskun et al. [55] found a correlation between Fok-I polymorphism and serum 25(OH)D_3_ levels in ASD patients. Also Xia et al. [57] proved that vitamin D receptor (Bsm-I, Apa-I, and Taq-I) mutations and lower 25(OH)D_3_ levels are associated with Crohn’s disease in Chinese patients. Few other studies indicated no difference in the level of 25(OH)D_3_ in serum and VDR polymorphisms in children with atopic dermatitis [42], prostate cancer [58], and multiple sclerosis [59].

Vitamin D_3_ plays a crucial role in its impact on the development and function of the brain [60], and vitamin D is therefore implicated in neuropsychiatric disorders, such as autism spectrum disorder [11]. By interaction with the specific VDR, the developmental and functional consequences of vitamin D in the nervous system can be modulated. It was shown that patients bearing mutations in their VDR receptor gene might have a different activation threshold than the wild form of the receptor [61,62]. It was also shown that polymorphisms in the VDR gene could modulate bone mineral density [63], and cause inactivation of vitamin D receptor effects on skeletal and calcium homeostasis [64]. As all molecular actions of 1,25(OH)_2_D_3_ are mediated by the VDR [65] the presence of SNPs can influence the gene expression [66], and thus genetic polymorphism in the VDR receptor gene may be associated with altered VDR expression and function in the presence of functional 1,25(OH)_2_D_3_ rather than with the serum level of vitamin D_3_ or its precursors. As vitamin D_3_ deficiency occurs very often in pregnant women [66,67,68,69], and vitamin D plays a crucial role in neuropsychiatric disorders, the importance of vitamin D_3_ in the development of autism is implicated [11]. Moreover, because 90% of human vitamin D_3_ stores come from skin production [70], it is also suggested [11,71] that the autism increasing rates are linked with sun-avoidance programs for pregnant women, newborns and infants. Whitehouse et al. [15] showed that offspring of mothers with low vitamin D_3_ concentration (<49 nmol/L) were linked with an increased risk for development of autism. Similar results were presented by Humble et al. [72], who showed below-normal (31.5–40 nmol/L) concentrations of vitamin D in serum of patients with autism, schizophrenia and ADHD. The relevance of epigenetic programming following dietary supplementation with vitamin D in the mother during pregnancy, which can induce a persistent functional epigenetic state in the newborn thereby contributing to a decreased susceptibility to develop ASD despite its genetic risk, warrants further study of VDR as an ASD candidate gene in larger cohorts.

In recent years, autism has grown to an “epidemic” with a 50-fold increase in prevalence during the last 25 years [73], and it is important to search for new methods of diagnosis and prevention. Genetic alterations of the VDR gene could lead to defects in gene activation and functioning of the receptor protein, and the detection of genetic polymorphisms in the VDR gene driving disease susceptibility can be a useful instrument in preventive medicine [21]. Many conflicting studies were published on the impact of vitamin D deficiency on the development of particular diseases, including autism, and therefore, the attention should be focused on genes that control vitamin D metabolism and efficient functioning in the body. Indeed, the final outcome is a combination of an environmental factor (vitamin D) and genetic background (mutations of VDR gene) and is apparent in many tissues as the VDR is widely expressed.

In conclusion, this study proved genetic polymorphism in two SNP in VDR may be correlated with the development of ASD symptoms. Since vitamin D_3_ serum levels were not significantly different between ASD and non-ASD children this could indicate that the functionality of the vitamin D_3_ metabolism might be affected when contributing to the development of symptoms among ASD children.

## Figures and Tables

**Figure 1 brainsci-07-00115-f001:**
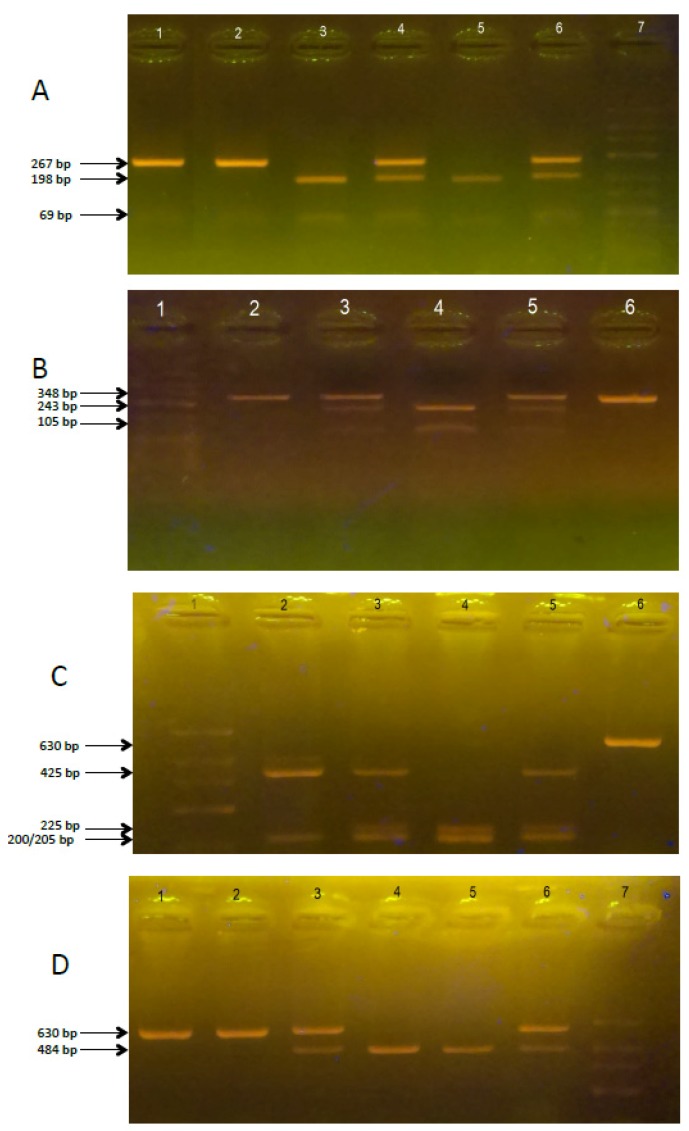
The electrophoregram of VDR receptor gene fragments genotyping. (**A**)—Fok-I electrophoregram: Path 1—PCR product (267 bp), path 2—FF homozygote (267 bp), path 3 and 5—ff homozygote (198, 69 bp), path 4 and 6—heterozygous Ff (267, 198, 69 bp), path 7—molecular marker; (**B**)—Bsm-I electrophoregram: Path 1—molecular marker, path 2—BB homozygote (348 bp), path 3 and 5—heterozygous Bb (348, 243, 105 bp), path 4 and 6—bb homozygote (243, 105 bp), path 6—PCR product (348 bp); (**C**)—Taq-I electrophoregram: Path 1—molecular marker, path 2—TT homozygote (425, 205 bp), path 3 and 5—heterozygous Tt (425, 225, 205, 200 bp), path 4—tt homozygote (225, 205, 200 bp), path 5—PCR product (630 bp); (**D**)—Apa-I electrophoregram: Path 1—PCR product (630 bp), path 2—AA homozygote (630 bp), path 4 and 5—aa homozygote (484, 146 bp), path 3 and 6—heterozygous Aa (630, 484, 146 bp), path 7—molecular marker. The DNA band of 146 bp size is not visible on the gel.

**Table 1 brainsci-07-00115-t001:** Primers for VDR SNPs and PCR-RFLP conditions.

SNP	Primer Sequence	Restriction Enzyme	PCR/RFLP Products (bp)
Fok-I	Fok1R: 5-ATGGAAACACCTTGCTTCTTCTCCCTC-3 Fok11F: 5-AGCTGGCCCTGGCACTGACTCtGGCTCT-3	FastDigest Fok-I	ff: 198, 69 FF: 267 Ff: 267, 198, 69 PCR product: 267
Bsm-I	ABsm1F: 5-CGGGGAGTATGAAGGACAAA-3 ABsm1R: 5-CCATCTCTCAGGCTCCAAAG-3	FastDigest Bsm-I	bb: 243, 105 BB: 348 Bb: 348, 243, 105 PCR product: 348
Taq-I	ATaq1F: 5-GGATCCTAAATGCACGGAGA-3 ATaq1R: 5-AGGAAAGGGGTTAGGTTGGA-3	FastDigest Taq-I	tt: 225, 200, 205 TT: 425, 205 Tt: 425, 225, 200, 205 PCR product: 630
Apa-I	ATaq1F: 5-GGATCCTAAATGCACGGAGA-3 ATaq1R: 5-AGGAAAGGGGTTAGGTTGGA-3	FastDigest Apa-I	aa: 484, 146 AA: 630 Aa: 630, 484, 146 PCR product: 630

**Table 2 brainsci-07-00115-t002:** Genotype and allele frequencies of VDR SNPs and associations with autism.

Genotype/Allele	ASD *n* (%)	Control *n* (%)	OR (95% CI) Control vs. ASD	*p*-Value
Fok-I (rs2228570) FF	38 (35)	68 (35)		
Ff	51 (47)	92 (47)	0.99 (0.59–1.68)	0.98
ff	19 (18)	36 (18)	0.94 (0.48–1.87)	0.87
F	127 (58.8)	228 (58.2)	0.97 (0.7–1.4)	0.88
f	89 (41.2)	164 (41.8)		
FF vs. Ff + ff	0.97 (0.66–1.42)			0.88
Bsm-I (rs1544410)				
BB	12 (11)	24 (12)		
Bb	67 (62)	118 (60)	1.14 (0.53–2.42)	0.74
bb	29 (27)	54 (28)	1.07 (0.47–2.46)	0.86
B	91 (42.1)	166 (42.3)	1.01 (0.72–1.41)	0.96
b	125 (57.9)	226 (57.7)		
BB vs. Bb + bb	1.11 (0.65–1.9)			0.71
Taq-I (rs731236) TT	33 (31)	92 (47)		
Tt	61 (56)	85 (43)	2.00 (1.19–3.35)	0.008
tt	14 (13)	19 (10)	2.05 (0.93–4.56)	0.08
T	127 (58.8)	269 (68.6)	1.53 (1.08–2.16)	0.02
t	89 (41.2)	123 (31.4)		
TT vs. Tt + TT	2.08 (1.40–3.01)			0.0004
Apa-I (rs7975232)				
	22 (20)	26 (13)	3.32 (1.44–7.63)	0.004
Aa	73 (68)	119 (61)	2.41 (1.23–4.73)	0.01
aa	13 (12)	51 (26)		
A	117 (54.2)	171 (43.6)	1.53 (1.09–2.13)	0.01
a	99 (45.8)	221 (56.4)		
aa vs. AA + Aa	2.65 (1.6–4.3)			0.0001

Genotype frequencies of VDR SNPs were determined in the control group and patients with autism, and also associations with autism. Odds ratios (ORs) with 95% confidence interval (CI) and *p* values were calculated for the wild/wild genotype versus the wild/mutant and mutant/mutant genotypes.

**Table 3 brainsci-07-00115-t003:** Genotype frequencies of VDR SNPs and associations with gender.

	ASD + non-ASD Control		OR (95% CI) Females vs. Males	*p*-Value
	Female *n* (%)	Male *n* (%)		
Fok-I (rs2228570)				
FF	39 (34)	67 (36)	FF vs. Ff + ff	0.61
Ff	44 (47)	88 (47)	0.91 (0.62–1.32)	
Ff	22 (19)	33 (17)		
Bsm-I (rs1544410)				
BB	13 (11)	23 (12)	BB vs. Bb + bb	
Bb	67 (58)	118 (63)	0.9 (0.53–1.52)	0.69
Bb	35 (30)	48 (25)		
Taq-I (rs731236)				
TT	57 (50)	68 (36)	TT vs. Tt +.tt	
Tt	47 (41)	99 (52)	1.76 (1.2–2.6)	0.0035
Tt	11 (9)	22 (10)		
Apa-I (rs7975232)				
AA	11 (9)	37 (20)	aa vs. AA + Aa	
Aa	80 (70)	112 (59)	1.07 (1.07–1.65)	0.75
Aa	24 (21)	40 (21)		

Genotype frequencies of VDR SNPs were determined in the non-ASD control group and patients with autism, and also associations with gender. Odds ratios (ORs) with 95% confidence interval (CI) and *p* values were calculated for the wild/wild genotype versus the wild/mutant and mutant/mutant genotypes.

**Table 4 brainsci-07-00115-t004:** Vitamin D_3_ concentration depending on VDR receptor polymorphisms (Apa-I, Taq-I, Bsm-I and Fok-I) in ASD patients.

VDR Gene Polymorphism	No. of Patients	Concentration (ng/mL)	SEM	*p*-Value
Apa-I				
Aa	9	41.6 (21.2–71.1)	13.4	0.07
AA	10	57.2 (19.9–72.5)	15.1	
Taq-I				
TT	4	48.8 (36.3–71.1)	15.8	0.71
Tt	11	48.5 (19.9–72.5)	14.7	
Tt	4	52.3 (21.2–71.0)	17.1	
Bsm-I				
BB	5	48 (26.2–69.4)	12.2	0.92
Bb	7	54.9 (21.2–72.5)	21.2	
Bb	7	52.4 (31.3–72.2)	18.8	
Fok-I				
FF	7	52.7 (31.3–72.5)	13.1	0.21
Ff	8	50.3 (21.2–72.2)	10.6	
ff	4	30.6 (19.9–45.5)	11.7

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
