# Peer review of "Vitamin D Receptor Gene Polymorphisms Associated with Childhood Autism"

_brainsci, 2017, doi:10.3390/brainsci7090115_

Round 1

Reviewer 1 Report

This study examined the distribution of several vitamin D receptor (VDR) SNPs among autistic subjects in Poland vs neurotypical control subjects. The study appears to be well-conducted, with reliable results. Significant variations were observed for several loci in a gender-specific manner. The results support a role for VDR genetic variants as a risk factor for autism.

1. VDR SNPs are identified via their restriction enzyme sensitivity, which is typical of the field, but this designation does not allow appreciation of the specific nucleotide changes. This information might be important in interpreting the results. Are they exonic or intronic? Do they involve CpG loci? Do they affect the VDR amino acid sequence? Moreover, references and discussion about the functional effect of specific VDR SNPs are missing. This missing information would facilitate interpretation of the interesting findings, especially in the context of autism.

2. The authors mention the potential contribution of epigenetic mechanisms to autism, but do not specifically relate this to VDR SNPs. Antioxidant status regulates DNA methylation, and is impaired in autism and VDR regulates two enzymes in the transsulfuration pathway (CBS and CSE) leading to synthesis of the antioxidant glutathione (GSH). This would connect the current results to epigenetic dysfunction and autism. A further connection arises from VDR regulation of parvalbumin expression, since parvalbumin-expressing interneurons are important for attention and neuronal synchronization, and their density is lower in autism.

3. Figure 2 is useless and adds nothing to the paper. It could be found in any textbook. Certainly autism should be brought into any summary.

4. In the Materials and Methods section there is no mention of exclusion of subjects taking vitamin D supplements.

5. The influence of gender within the autism cohort was apparently not explored, but should be reported, in addition to the combined cohorts in Table 3. 

6. Haplotype analysis might have added further interest.

7. Grammar in the paragraph beginning on line 227 needs to be improved.

8. Line 64: reviews, not revievs    

Author Response

Reviewer 2

Comments and Suggestions for Authors

This study examined the distribution of several vitamin D receptor (VDR) SNPs among autistic subjects in Poland vs neurotypical control subjects. The study appears to be well-conducted, with reliable results. Significant variations were observed for several loci in a gender-specific manner. The results support a role for VDR genetic variants as a risk factor for autism.

1. VDR SNPs are identified via their restriction enzyme sensitivity, which is typical of the field, but this designation does not allow appreciation of the specific nucleotide changes. This information might be important in interpreting the results. Are they exonic or intronic? Do they involve CpG loci? Do they affect the VDR amino acid sequence? Moreover, references and discussion about the functional effect of specific VDR SNPs are missing. This missing information would facilitate interpretation of the interesting findings, especially in the context of autism.

ANSWER: Polymorphisms Apa-I and Bsm-I  were found in intron 8. Taq-I was observed in exon 9, as a silent mutation, missense Fok-I polymorphism in exon 2, and this information is included in the Introduction section. These 4 SNPs do not affect the VDR amino acid sequence. This information is mentioned at the end of Introduction part. We did not focus on the SNP effect itself, but we cited few reviews with this information. We have indicated this more clearly in the manuscript.

2. The authors mention the potential contribution of epigenetic mechanisms to autism, but do not specifically relate this to VDR SNPs. Antioxidant status regulates DNA methylation, and is impaired in autism and VDR regulates two enzymes in the transsulfuration pathway (CBS and CSE) leading to synthesis of the antioxidant glutathione (GSH). This would connect the current results to epigenetic dysfunction and autism. A further connection arises from VDR regulation of parvalbumin expression, since parvalbumin-expressing interneurons are important for attention and neuronal synchronization, and their density is lower in autism.

ANSWER: In this study, we focused only on the relationship of SNP to autism and correlation with vitamin D3 thereby avoiding the role of methylation in this process. We agree with the reviewer about the importance of methylation but consider this a separate study worth of continuation, but currently out of scope of this manuscript.

3. Figure 2 is useless and adds nothing to the paper. It could be found in any textbook. Certainly autism should be brought into any summary.

ANSWER: we agree with the reviewer and have deleted this figure.

4. In the Materials and Methods section there is no mention of exclusion of subjects taking vitamin D supplements.

ANSWER: We agree with the reviewer and have now added explicitly that we included only children without vitamin D3 supplementation.

5. The influence of gender within the autism cohort was apparently not explored, but should be reported, in addition to the combined cohorts in Table 3. 

ANSWER: As stated in 2.1 the cohort of 108 ASD children contained 91 male and 17 female children. We considered the number of female ASD children too low to allow us to separately study the influence of gender. We agree with the reviewer that this is an interesting question, but this would require a greatly expanded ASD population of children beyond our current possibilities and thus beyond the scope of this manuscript.

6. Haplotype analysis might have added further interest.

ANSWER: We tested haplotypes, but because of the small sample size we did not further expand on these data in the manuscript. Please refer also to our answer to question 8 of reviewer 1: We tested genotypes for Apa and Taq together, and also for other loci. But because of the relatively small sample size we did not expand the data with pairs of loci. We now agree with the reviewer to present the 45% with Aa/Tt genotype in ASD as haplotypes (also in line with comment from reviewer 2). In ASD we reported 36% TA, 30% Ta, 20% tA and 14% ta, while in the non-ASD children we found 34% TA, 42% Ta, 8% tA and 16% ta. Therefore, our results suggest that tA is 2.5x more frequent in ASD children than in non-ASD children, despite the fact that these data need now to be confirmed in larger groups. We have now changed this in the manuscript.

7. Grammar in the paragraph beginning on line 227 needs to be improved.

ANSWER: we agree and have changed the sentence.

8. Line 64: reviews, not revievs   

ANSWER: we agree and have corrected the mistake. 

Reviewer 2 Report

The manuscript of Cieslinska et al. describes study of several previously described polymorphisms in the vitamin D receptor (VDR) gene of humans.  Specifically, the authors quantify the presence or absence of particular SNPs in children with autistic spectrum disorder (ASD) versus children without ASD.  Vitamin D is known to be essential to proper brain development, and thus mutations in VDR could ostensibly lead to a wide range of developmental/cognitive defects such as those seen in ASD patients.  The principal discovery of the research is that SNPs at two different locations within the VDR are correlated with the absence of ASD symptoms (or stated another way, there are allelic differences that may be responsible for a predisposition, or lack thereof, to VDR).  The authors did not see significant differences in vitamin D levels in the blood of ASD and control children.

The importance of the author's discoveries is somewhat unclear.  While VDR may be involved in the progression of ASD, it does not appear to be the "smoking gun" behind the development of autism.

I have a number of suggestions and questions.  Many of these are included on a PDF markup of the manuscript.  As there is no apparent means of attaching such a file directly to my review, I will submit it to the editor in hopes that he/she will forward it to the authors.  Some of the main questions/issues are noted below.

(1) The conclusion in the abstract suggests that vitamin D3 concentration has been ruled out as a potential cause of ASD.  This is misleading.  I suppose that the authors mean to say that since vitamin D3 concentrations were not significantly different among control and ASD children, their findings of correlations between two VSD SNPs and occurrence of ASD is not simply due to the fact that the ASD children had low vitamin D levels.  It appears to have a real genetic basis.  Is this true?

(2) I would avoid referring to the control children as "healthy children."  What constitutes a healthy child is open to much debate.  I would refer to the control group as children with no known behavioral disorders or non-ASD children.

(3) In Figure 1, the authors refer to their gel images as electropherograms.  These are NOT electropherograms (please Google the word "electropherogram").

(4) Section 2.3 is horribly written and lacks detail.  Is the vitamin D3 source whole blood or plasma?  Do the ELISA plates contain anti vitamin-D3 conjugated to horseradish peroxidase?  It sounds like the horseradish peroxidase was added to each reaction.  If so, what is conjugated to the antibodies?  Is the same volume of liquid placed in each microtiter plate well?  How much blood/serum was used to determine vitamin D3 concentration?

(5) In section 2.4, the authors say, "Only all 4-genotyped SNPs probes were used in the calculations."  What does this mean?  Would we expect other probes to be assayed?

(6) The alleles are given names like they are Mendelian genes with dominant and recessive versions (e.g., FF, Ff, ff).  I assume that these names are arbitrary and are not meant to be suggest dominance or lack thereof.  In any case, the upper and lower case designations need to be explained to prevent confusion.

(7) The acronyms in Table 2, etc., need to be defined.  I was able to look up what OR and CI are, but they should be spelled out in the text before they appear as acronyms.

(8) The authors show that T and a alleles are associated with a decreased risk of ASD.  However, it appears that no attempt was made to look at the frequency of pairs of "loci" in children (for example, did ASD children more frequently have a t and an A allele together than non-ASD children?  Were ttAA genotypes more likely to have ASD than other combinations?  It seems this should have been examined at least for the Taq-1 and Apa-1 positions?  There may be good reasons the authors didn't explore this question, but if so, it would be useful to know why.

(9) As in the abstract, the conclusion statement at the end of the paper is confusing.

Author Response

Reviewer 1

Comments and Suggestions for Authors:

The manuscript of Cieslinska et al. describes study of several previously described polymorphisms in the vitamin D receptor (VDR) gene of humans.  Specifically, the authors quantify the presence or absence of particular SNPs in children with autistic spectrum disorder (ASD) versus children without ASD.  Vitamin D is known to be essential to proper brain development, and thus mutations in VDR could ostensibly lead to a wide range of developmental/cognitive defects such as those seen in ASD patients.  The principal discovery of the research is that SNPs at two different locations within the VDR are correlated with the absence of ASD symptoms (or stated another way, there are allelic differences that may be responsible for a predisposition, or lack thereof, to VDR).  The authors did not see significant differences in vitamin D levels in the blood of ASD and control children.

The importance of the author's discoveries is somewhat unclear.  While VDR may be involved in the progression of ASD, it does not appear to be the "smoking gun" behind the development of autism.

I have a number of suggestions and questions.  Many of these are included on a PDF markup of the manuscript.  As there is no apparent means of attaching such a file directly to my review, I will submit it to the editor in hopes that he/she will forward it to the authors.  Some of the main questions/issues are noted below.

(1) The conclusion in the abstract suggests that vitamin D3 concentration has been ruled out as a potential cause of ASD.  This is misleading.  I suppose that the authors mean to say that since vitamin D3 concentrations were not significantly different among control and ASD children, their findings of correlations between two VSD SNPs and occurrence of ASD is not simply due to the fact that the ASD children had low vitamin D levels.  It appears to have a real genetic basis.  Is this true?

ANSWER:

Yes, we suggest that genetic polymorphism may be correlated with autism.

We also suggest that:

1. vitamin D3 concentrations were not significantly different among control and ASD children

2. Two VDR polymorphisms are more common in ASD children

3. Despite vit.D3 concentration is not directly correlated with autism, VDR polymorphism may influence on functionality of vit.D3 metabolites on the body and correlate with autism (vitamin D receptor may not work correctly) .

We have changed the conclusion in the Abstract and at the end of the manuscript accordingly.

(2) I would avoid referring to the control children as "healthy children."  What constitutes a healthy child is open to much debate.  I would refer to the control group as children with no known behavioral disorders or non-ASD children.

ANSWER: We agree with the reviewer and have corrected this throughout the manuscript.

(3) In Figure 1, the authors refer to their gel images as electropherograms.  These are NOT electropherograms (please Google the word "electropherogram").

ANSWER: we agree with the reviewer about the letter-mistake, and have changed this into “electrophoregram”.

(4) Section 2.3 is horribly written and lacks detail.  Is the vitamin D3 source whole blood or plasma?  Do the ELISA plates contain anti vitamin-D3 conjugated to horseradish peroxidase?  It sounds like the horseradish peroxidase was added to each reaction.  If so, what is conjugated to the antibodies?  Is the same volume of liquid placed in each microtiter plate well?  How much blood/serum was used to determine vitamin D3 concentration?

ANSWER: We analyzed serum for vitamin D content, according to the manufacturer’s instruction (QAYEE-BIO), with a Human Vitamin D3 ELISA kit suitable to be used with serum and plasma. The kit used is a double-antibody (two different 25-dihydroxyvitamin D-specific antibodies) sandwich ELISA with the detecting antibody conjugated to horse-radish peroxidase. Thus, the horseradish peroxidase-labeled antibody was added as a conjugate. The same volume of liquid was placed in each microtiter plate well in all steps throughout the procedure. In addition, 50 µl of serum was used to determine the vitamin D3 concentration. We have rewritten this section containing now this information.

(5) In section 2.4, the authors say, "Only all 4-genotyped SNPs probes were used in the calculations."  What does this mean?  Would we expect other probes to be assayed?

ANSWER: We started our tests with larger groups than the 108 ASD and 196 non-ASD children. However, some DNA in the samples was damaged and could not be processed further and therefore, we selected only those children that enabled us to type all four selected SNPs at the same time. We did not separately analyzed 2 or 3 SNPs as not enough material was available. We agree with the reviewer and have removed this sentence to avoid any confusion.

(6) The alleles are given names like they are Mendelian genes with dominant and recessive versions (e.g., FF, Ff, ff).  I assume that these names are arbitrary and are not meant to be suggest dominance or lack thereof.  In any case, the upper and lower case designations need to be explained to prevent confusion.

ANSWER: Names of alleles are widely used and taken from literature (references 21,32,33,34,45) and they do not suggest any dominance. We have clarified this in the manuscript.

(7) The acronyms in Table 2, etc., need to be defined.  I was able to look up what OR and CI are, but they should be spelled out in the text before they appear as acronyms.

ANSWER: We have corrected this in the manuscript: OR is odds ratio and CI is confidence interval.

(8) The authors show that T and a alleles are associated with a decreased risk of ASD.  However, it appears that no attempt was made to look at the frequency of pairs of "loci" in children (for example, did ASD children more frequently have a t and an A allele together than non-ASD children?  Were ttAA genotypes more likely to have ASD than other combinations?  It seems this should have been examined at least for the Taq-1 and Apa-1 positions?  There may be good reasons the authors didn't explore this question, but if so, it would be useful to know why.

ANSWER: We tested genotypes for Apa and Taq together, and also for other loci. But because of the relatively small sample size we did not expand the data with pairs of loci. We now agree with the reviewer to present the 45% with Aa/Tt genotype in ASD as haplotypes (also in line with comment from reviewer 2). In ASD we reported 36% TA, 30% Ta, 20% tA and 14% ta, while in the non-ASD children we found 34% TA, 42% Ta, 8% tA and 16% ta. Therefore, our results suggest that tA is 2.5x more frequent in ASD children than in non-ASD children, despite the fact that these data need now to be confirmed in larger groups. We have now changed this in the manuscript.

(9) As in the abstract, the conclusion statement at the end of the paper is confusing.

ANSWER: We have clarified the conclusion in the manuscript.